# VoCAPTER: Voting-based Pose Tracking for Category-level Articulated Object via Inter-frame Priors

**Li Zhang**
Hefei Institute of Physical Science, Chinese Academy of Sciences
University of Science and Technology of China
Astribot Inc
HeFei, AnHui, China
zanly@mail.ustc.edu.cn

**Zean Han***
Department of Mathematics, Chinese University of Hong Kong, New Territories, Hong Kong SAR
HongKong, China
zhan@math.cuhk.edu.hk

**Yan Zhong**
School of Mathematical Sciences, National Engineering Research Center of Visual Technology, Peking University.
Beijing, China
zhongyan@stu.pku.edu.cn

**Qiaojun Yu**
Department of Computer Science, Shanghai Jiao Tong University
ShangHai, China
yqjllxs@sjtu.edu.cn

**Xingyu Wu**
The Hong Kong Polytechnic University Hong Kong SAR
HongKong, China
xingy.wu@polyu.edu.hk

**Xue Wang**[†]
Hefei Institute of Physical Science, Chinese Academy of Sciences
HeFei, AnHui, China
xwang@iim.ac.cn

**Rujing Wang**[†*]
Hefei Institute of Physical Science, Chinese Academy of Sciences
HeFei, AnHui, China
rjwang@iim.ac.cn

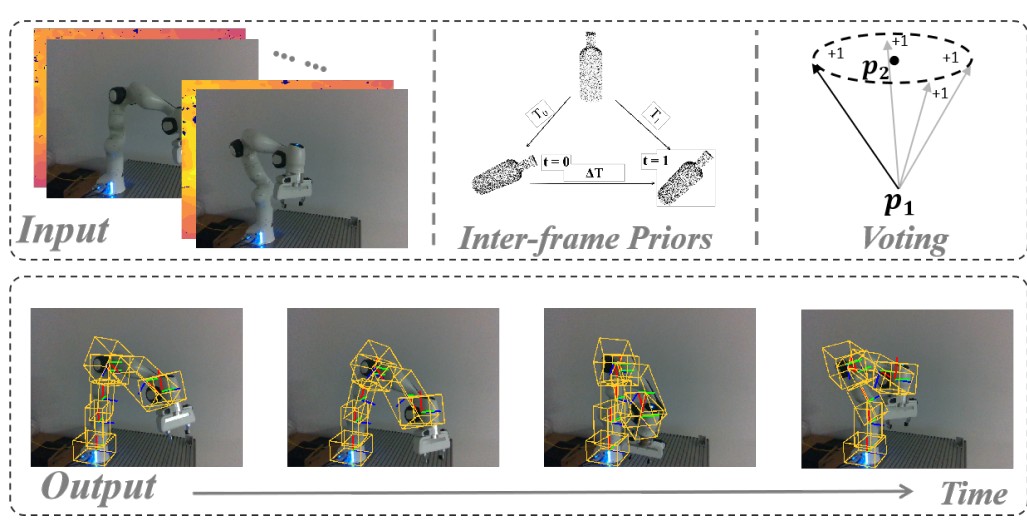

**Figure 1: Given a live point cloud stream or RGB-D images as input, our VoCAPTER can track the per-part pose at all frames $t > 0$ in an online manner. Our main contributions are: 1) we turn the articulated object pose tracking task into an inter-frame pose increment estimation task instead of tracking frame by frame. 2) We leverage the SE(3)-invariant parameters to conduct per-part pose voting instead of direct pose regress. Experiments demonstrate the superiority of our VoCAPTER, not only on the synthetic dataset but also on real-world scenarios.**

*Li Zhang and Zean Han contributed equally. This work was done when Li Zhang was an intern at Astribot Inc. Corresponding authors: Xue Wang and Rujing Wang.

## Abstract

Articulated objects are common in our daily life. However, current category-level articulation pose works mostly focus on predicting 9D poses on statistical point cloud observations. In this paper, we deal with the problem of category-level online robust 9D pose tracking of articulated objects, where we propose **VoCAPTER**, a novel 3D **Vo**ting-based **C**ategory-level **A**rticulated object **P**ose Track**ER**. Our VoCAPTER efficiently updates poses between adjacent frames by utilizing partial observations from the current frame and the estimated per-part 9D poses from the previous frame. Specifically, by incorporating prior knowledge of continuous motion relationships between frames, we begin by canonicalizing the input point cloud, casting the pose tracking task as an inter-frame pose increment estimation challenge. Subsequently, to obtain a robust pose-tracking algorithm, our main idea is to leverage SE(3)-invariant features during motion. This is achieved through a voting-based articulation tracking algorithm, which identifies keyframes as reference states for accurate pose updating throughout the entire video sequence. We evaluate the performance of VoCAPTER in the synthetic dataset and real-world scenarios, which demonstrates VoCAPTER's generalization ability to diverse and complicated scenes. Through these experiments, we provide evidence of VoCAPTER's superiority and robustness in multi-frame pose tracking of articulated objects. We believe that this work can facilitate the progress of various fields, including robotics, embodied intelligence, and augmented reality. All the codes will be made publicly available.

## CCS Concepts

• **Computing methodologies → Vision for robotics**.

## Keywords

Voting, Inter-frame Priors, Category-level Objects, Pose Tracking

**ACM Reference Format:**
Li Zhang, Zean Han, Yan Zhong, Qiaojun Yu, Xingyu Wu, Xue Wang, and Rujing Wang. 2024. VoCAPTER: Voting-based Pose Tracking for Category-level Articulated Object via Inter-frame Priors. In *Proceedings of Make sure to enter the correct conference title from your rights confirmation emai (ACM MM'24)*. ACM, Melbourne, Australia, 10 pages. https://doi.org/XXXXXXX. XXXXXXX

## 1 INTRODUCTION

Articulated objects are omnipresent in human daily life. Effectively conducting many downstream tasks relies heavily on articulated object pose estimation and tracking, encompassing human-object interaction [16, 17, 25, 43], robotic manipulation [4, 47], augmented reality [2, 3, 6, 7], and 3D scene understanding [1, 5, 9]. Unlike merely predicting *single-frame* articulated object poses from statistical point clouds or RGB-D images [15, 36, 40], articulated object pose tracking tasks necessitate handling *multi-frame* point cloud sequences, thus amplifying challenges. Specifically, point cloud sequences furnish temporal continuity information of articulated objects, hence demanding consideration of pose variations over time. Objects may undergo rotations and translations between different frames, therefore increasing task complexity due to this dynamic nature.

To deal with category-level articulated object 9D pose tracking, a straightforward idea is to migrate the single-frame articulation pose estimation methods (such as keypoint based DAKDN [45] and reinforcement learning based ArtPERL [21]) into the multi-frame pose tracking task. However, these solutions face two main challenges:

**(i) Tracking Manner.** The efficacy of these methods is hampered by the need for per-pixel representation learning, which consequently impacts the tracking speed in live point cloud streams. Consequently, these constraints obstruct the ability of category-level articulated object trackers to attain satisfactory robust and real-time tracking performance.

**(ii) Pose Modeling Problem.** Objects have their inherent structures, which are invariant to rotations and translations (*e.g.*, the relative position of the handle does not change during the rotation of the mug). However, prior arts often directly regress 9D poses without accounting for these intrinsic properties. This oversight results in ill-posed pose modeling issues, making convergence challenging and compromising the retention of useful features.

To address the first challenge, our core idea involves leveraging the prior information between adjacent frames. While single-frame pose estimation methods can be directly applied to pose tracking tasks, this approach essentially predicts frame by frame, overlooking crucial prior information. In other words, previous methods fail to adequately consider the continuous motion relationships between frames, which hinders the improvement of inference speed. Secondly, to alleviate the difficulty of predicting point cloud poses observed in camera space, our strategy involves using the inverse pose transformation of the previous frame's point cloud to normalize the input space of the current frame. Based on this strategy, we transform the articulated object pose tracking task into an inter-frame pose increment estimation task to better utilize the temporal coherence and structural consistency of consecutive frames in video sequences, thereby improving the network's inference speed and accuracy.

To handle the second challenge, the basic mechanism behind our method is to conduct a voting-based prediction, which takes the SE(3)-invariant features into consideration. The key idea is to turn the geometric features into the probability features on the point cloud, which is more friendly for learning. Concretely, we first globally model each rigid part's structure by defining key geometric relationships (*i.e.*, some SE(3)-invariant parameters). Then we seek local geometric features by adopting a local matching strategy where each part independently casts *votes* for its estimated pose based on local features from point pairs and the global structure from the corresponding part. Furthermore, to get a robust prediction, we additionally introduce a part awareness mechanism to assign scores for the candidate point pairs instead of the noisy data that participates in voting.

We evaluate our VoCAPTER on category-level pose tracking tasks on both point clouds dataset (PartNet-Mobility) as well as an RGB-D images dataset (ReArt-48). Experiments on real-world scenarios (RobotArm) also demonstrate the generalization capacity of our VoCAPTER. Our contributions can be summarized as follows:

- We tailor a novel **Vo**ting-based **C**ategory-level **A**rticulated object **P**ose Track**ER** (**VoCAPTER**) to break the performance bottleneck of existing methods.

- We turn the articulated object pose tracking task into an inter-frame pose increment estimation task via the inter-frame priors. Besides, a universal voting strategy is applied to track the pose frame by frame to avoid the weakness of direct pose regression methods.
- The efficiency and robustness of the VoCAPTER are demonstrated through the evaluation of either synthetic or real-world scenarios. The experimental results show the dramatic performance improvement and generalization capacity of our method.

## 2 RELATED WORK

### 2.1 Category-level Articulation Pose Estimation

Category-level object pose estimation involves predicting the pose of objects that may not have been encountered previously [8, 23, 27, 35]. Unlike rigid objects[11, 28, 29], articulated objects present a more intricate challenge due to the interdependence and coordinated movement of multiple joints [30, 32, 40]. It surely adds difficulty to the pose estimation task, as it requires considering the constrained relationships and coordinated movements among the joints in computer vision tasks [33 ? ]. Addressing the challenge of category-level pose estimation for articulated objects, A-NCSH [18] pioneers the extension of Normalized Object Coordinate Space (NOCS) [35] to articulate structures, enabling the estimation of part-level poses. To go further, Liu *et al.* [24] advance the approach towards real-world articulated object analysis by introducing part pairs to investigate previously unseen instances. Additionally, Xue *et al.* [42] propose a novel approach utilizing key-points as articulation models, aiming to speed up the inference while maintaining accurate pose estimation.

Although the aforementioned studies effectively address category-level articulation pose estimation with satisfactory performance, their applicability as ready-to-use solutions in pose tracking tasks is limited. This limitation arises from the dense prediction paradigm, which constrains both the robustness and the inference speed of these methods.

### 2.2 Category-level Articulation Pose Tracking

Unlike traditional articulated object pose estimation tasks [8, 12, 35], category-level articulation pose tracking [26, 34, 39] extends the scope to real-time point cloud streams. Specifically, this task aims to update the pose at the frame level, utilizing the depth point cloud of the current frame along with the estimated pose from the preceding frame. One notable approach in prior research, BundleTrack[37], leverages the complementary attributes of recent advances in deep learning for segmentation and robust feature extraction for both instance and category-level objects Pose Tracking. Alternatively, another set of methodologies aims at keypoint-based object representation and modeling. These approaches typically involve selecting appropriate key points based on the object's structural characteristics, analyzing their spatial positions and relationships, modeling the object's representation and shape, and ultimately inferring its attitude information. Examples of such techniques include tracking objects using predicted 2D keypoints in RGB sequences [20], incorporating depth information for tracking [13], and leveraging tracked articulated object poses for 3D reconstruction [38].

Despite significant progress, the 9D pose of an articulated object (comprising 3D translation, 3D rotation, and 3D scale) introduces additional degrees of freedom and complexity, posing challenges if a direct prediction is applied. Consequently, direct pose tracking methods often yield unstable results. Drawing inspiration from BeyondPPF [44], we use a universal voting strategy for articulated object pose tracking in this paper. Our method employs a voting-based strategy, integrated into an end-to-end pipeline, which dynamically updates the pose relative to previous frames for per-part pose tracking.

## 3 Notations and Problem Statement

In this paper, we address the task of tracking per-part 9D pose of articulated objects belonging to known categories. We build upon the category-level articulated object and part definitions introduced in A-NCSH [18] and assume a constant number of rigid parts and kinematic structures for all objects within the same category. The problem of pose tracking is defined as follows: the task takes the live point cloud stream $\{\mathcal{P}_t\}_{t \geq 0}$ as input, where $t$ denotes the $t$-th frame in the video. Each point cloud $\mathcal{P}_t$ consists of $K$ rigid parts, and $S^{(k)}$ represents the points of the $k$-th part. Along with each point cloud, there exists a per-part pose $T_t^{(k)} = [R_t^{(k)} \mid \mathbf{t}_t^{(k)}]_{k=1}^{K}$ at the $t$-th frame. Here, $R_t^{(k)} \in SO(3)$ represents the rotation and $\mathbf{t}_t^{(k)} \in \mathbb{R}^3$ represents the translation. Our target is to track the per-part pose $T_t^{(k)} = [R_t^{(k)} \mid \mathbf{t}_t^{(k)}]_{k=1}^{K}$ at all frames $t > 0$ in an online manner, given a live stream of point clouds $\{\mathcal{P}_t\}_{t \geq 0}$ containing the per-part pose $T_0^{(k)} = [R_0^{(k)} \mid \mathbf{t}_0^{(k)}]_{k=1}^{K}$ at the 0-th frame.

The pipeline of our VoCAPTER can be illustrated as follows: for $t$-th frame, our framework needs to track the per-part pose $T_t^{(k)}$ with the given estimated $T_{t-1}^{(k)}$ from the last frame and the point cloud $\mathcal{P}_t$. A change of pose consists of the increments in rotation $\Delta R_t^{(k)} \in SO(3)$ and translation $\Delta \mathbf{t}_t^{(k)} \in \mathbb{R}^3$. Then the absolute pose of any part can be retrieved by applying the increments of pose by Equation 1:

$$T_t^{(k)} = \Delta T_t^{(k)} \cdot T_{t-1}^{(k)} = \Delta T_t^{(k)} \cdot \Delta T_{t-1}^{(k)} \cdots T_0^{(k)} \tag{1}$$

To track the pose in the current frame, our key idea is to generate the SE(3)-invariant parameters $\mu_t^{(k)}, \nu_t^{(k)}, \alpha_t^{(k)}, \beta_t^{(k)}, \gamma_t^{(k)}$ and conduct the voting procedure based on these parameters (Details can be seen in Section 4.4). To this end, we can effectively predict the pose of the current frame with the assistance of the last frame.

## 4 METHOD

The overview of our VoCAPTER can be seen in Figure 3, in this section, we will introduce each module in detail. Formally, taking a 3D partial point cloud or an RGB-D image as input, the $t$-th frame point cloud is first canonicalized via the estimated 9D pose $T_{t-1}^k$ from the last frame, as discussed in Section 4.1. Then we use PointNet++ to conduct the segmentation at the per-part level, as elaborated in Section 4.2. We propose a part awareness mechanism to filter the noisy point pairs in Section 4.3. Finally, given the SE(3)-invariant parameters, we perform the pose voting frame by frame, with details in Section 4.4.

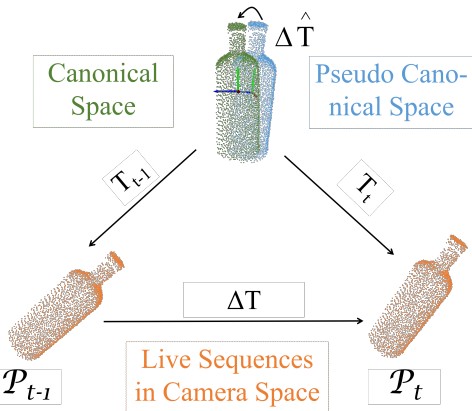

**Figure 2: The illustration of inter-frame priors implementation. Since we perform the pose tracking task at each rigid-part level, we choose a rigid object *bottle* to illustrate the process.**

## 4.1 Canonicalization

Canonicalization of the point cloud helps simplify the training and optimization of the network, which is widely used in previous arts [18, 21, 42]. However, It's impractical to canonicalize the current frame directly. To take full advantage of the priors between adjacent frames, we *pseudo-canonicalize* the point cloud at the $t$-th frame via the pose from the last frame. Here, we give the theorem with the corresponding proof that this all holds in Theorem 4.1:

THEOREM 4.1. *During the object motion, the transformation matrix between the adjacent frames is approximately equal to the transformation matrix from the canonical space to the pseudo-canonical space.*

PROOF. As depicted in Figure 2, denote the transformation matrix of the adjacent frames by $\Delta T$, whose action on the $t-1$-th frame point cloud $\mathcal{P}_{t-1}$ can transform the $t-1$-th frame point cloud into the $t$-th frame point cloud $\mathcal{P}_t$. By the assumption in the task of tracking, $\Delta T$ is approximately the identity matrix $I$, which means that the transformation matrix $T_{t-1}$: canonical space $\longrightarrow \mathcal{P}_{t-1}$ is approximately equal to the transformation matrix $T_t$: canonical space $\longrightarrow \mathcal{P}_t$. Since the action of $T_{t-1}$ on $\mathcal{P}_t$ can get the pseudo-canonical space and the action of $T_t$ on $\mathcal{P}_t$ can get the canonical space, the transformation matrix $\Delta \hat{T}$ from the canonical space to the pseudo-canonical space is approximately equal to the identity matrix $I$. Then there exists $\epsilon$, such that:

$$\|\Delta T - I\|_2 < \epsilon/2 \tag{2}$$

$$\|\Delta \hat{T} - I\|_2 < \epsilon/2 \tag{3}$$

Then,

$$\|\Delta T - \Delta \hat{T}\|_2 = \|\Delta T - I + I - \Delta \hat{T}\|_2 \tag{4}$$

$$\leq \|\Delta T - I\|_2 + \|I - \Delta \hat{T}\|_2 \tag{5}$$

$$\leq \epsilon/2 + \epsilon/2 = \epsilon \tag{6}$$

Here $\epsilon$ is a threshold for the approximation. Then we can draw the conclusion that $\Delta T$ is approximately equal to $\Delta \hat{T}$. □

Therefore, the foundational idea behind our canonicalization is as follows: firstly, we canonicalize the $t$-th frame point cloud $\{\mathcal{P}_t\}_{t\geq 0}$ with the per-part pose $T_t^{(k)}$ into the *pseudo canonical space* by using the per-part pose $T_{t-1}^{(k)}$ at the previous frame. In the articulation pose canonicalization process, the canonicalized point cloud $\hat{\mathcal{P}}_t$ is obtained by the action of the inverse transformation $T_{t-1}^{(k)}$ on $\mathcal{P}_t$. We conduct this procedure at the per-part level with the relative transformation matrix $\Delta T$ of the adjacent frames using the Equation 7:

$$\hat{\mathcal{P}}_t^{(k)} = (T_{t-1}^{(k)})^{-1} \mathcal{P}_t^{(k)} = (\Delta T_{t-1}^{(k)} \cdot \Delta T_{t-2}^{(k)} \cdots T_0^{(k)})^{-1} \cdot \mathcal{P}_0^{(k)} \tag{7}$$

As specified in Theorem 4.1, to predict the $\Delta T$ here, we can predict the $\Delta \hat{T}$ which transforms the canonical space to the pseudo-canonical space.

After the canonicalization at per-part level, we only need to predict the per-part pose increment $\Delta T_t^{(k)} = [\Delta R_t^{(k)} \mid \Delta \mathbf{t}_t^{(k)}]$ between $t$-th and $t-1$-th frame. Given the minimal rotation angle and translation between adjacent frames along the XYZ-axis, the neural network becomes more sensitive to subtle pose changes, thereby significantly improving tracking performance.

## 4.2 Part Segmentaion

Given the canonicalized point cloud $\mathcal{P}_t \in \mathbb{R}^{N\times 3}$ in Section 4.1, we use the PointNet++ [31] to extract the helpful feature $\mathcal{F}_1$. Once we get the features, we use a decoder to decouple $\mathcal{F}_1$ into $K$ masks $\left\{m_t^{(k)} \in \mathbb{R}^{N\times 1} \mid k = 1, ..., K\right\}$. Mathematically, the part-level point cloud can be retrieved by Equation 8:

$$\mathcal{P}_t^{(k)} = \mathcal{P}_t \cdot m_t^{(k)}, \text{ where } m_t^{(k)} = \mathbb{1}(S_t = k) \tag{8}$$

**Loss Function.** During the training, we adopt the MSE loss to conduct the part segmentation optimization:

$$\mathcal{L}_{seg} = \sum_{k=1}^{K} \|m_t^{(k)} - \hat{m}_t^{(k)}\|_2 \tag{9}$$

where $\hat{m}_t^{(k)}$ denotes the predicted mask and $m_t^{(k)}$ denotes as the GT mask.

## 4.3 Part Awareness

As mentioned before, our method needs to sample the point pairs to conduct the per-part pose tracking. Intuitively, only the point pair chosen from the same part is valid. Upon the consideration of such situation, we propose the part awareness mechanism to filter the noisy point pairs (point pairs that come from different parts) before the pose voting.

Concretely, denote all the sampled point pairs by $\mathcal{T}$, we first define a point pair score $\{c_j \mid j = 1, ..., J\}$ ($J$ is the number of sampled point pairs), which serves as a filter for noisy point pairs originating from different parts. Mathematically, given the part segmentation $\{\mathcal{P}_t^{(k)} \mid k = 1, ..., K\}$, point pair score $c_j$ can be formulated as Equation 10:

$$c_j = \begin{cases} 1, & \text{if } \mathcal{T}_j = (p_1, p_2) \in (\mathcal{P}_t^{(k)}, \mathcal{P}_t^{(k)}) \\ 0, & \text{otherwise} \end{cases} \tag{10}$$

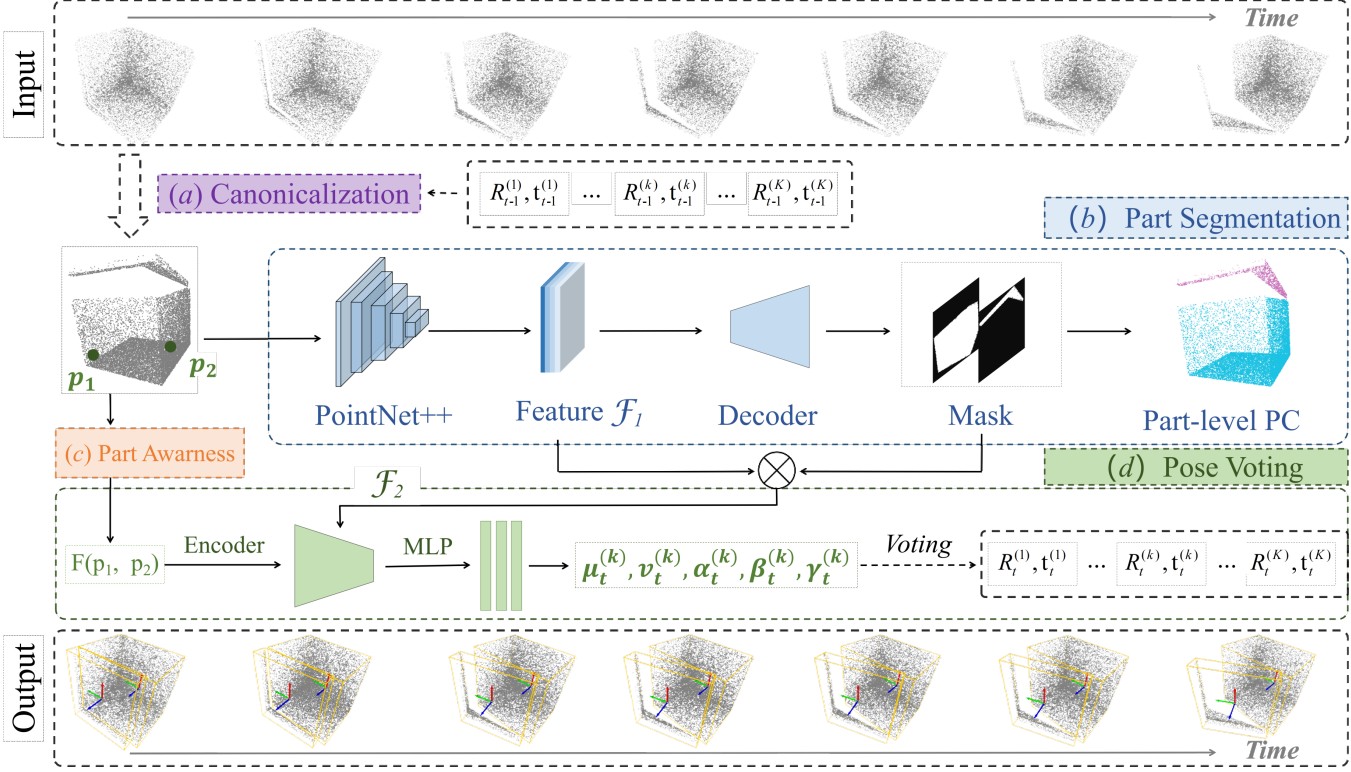

**Figure 3: The overview of our VoCAPTER framework. Taking the partial observation as input, our VoCAPTER consists of the following components: (1) Canonicalization. The canonicalized point cloud used for the downstream network. (Section 4.1) (2) Part Segmentation. We use the decomposed features to predict the $K$ Part Segmentation (Section 4.2). (3) Part Awareness is used to filter the noisy point pairs (Section 4.3). (4) Pose Voting. A voting-based strategy is used to conduct 9D pose tracking results (Section 4.4).**

During the inference stage, we retain only point pairs with scores $c_j$ greater than 0.5 to feed into the pose voting. Point pairs above this threshold are considered to be sampled from the same rigid part, effectively reducing interference from noisy point pairs.

**Loss Function.** The part awareness loss $\mathcal{L}_{part}$ is defined as the binary entropy loss of the ground truth $c_j$ and the predicted score $\hat{c}_j$:

$$\mathcal{L}_{part} = -\frac{1}{J}\sum_{j=1}^{J}\{c_j log(\hat{c}_j) + (1 - c_j)log(1 - \hat{c}_j)\}. \quad (11)$$

### 4.4 Pose Voting

For each rigid-part $\mathcal{P}_t^{(k)}$ at $t$-th frame, we sample the point pairs randomly from the same part as mentioned in Section 4.3. Subsequently, the point pair feature $F(p_1, p_2)$ [10] and the masked feature $\mathcal{F}_2$ are fed into the PPF encoder to get the re-modulated feature. Then, we use a three-layer MLP to output the SE(3)-invariant features, which is applied to generate the key parameters of voting $\mu_t^{(k)}, v_t^{(k)}, \alpha_t^{(k)}, \beta_t^{(k)}, \gamma_t^{(k)}$. We use the following strategy to obtain the $\Delta T_t^{(k)}$.

**Universal Voting Strategy.** In this section, we use a universal voting strategy that creates a global model description via randomly sampled oriented points. The essence of this method lies in leveraging feature sparsity. Concretely, pose tracking tasks for

non-uniformly distributed objects (*e.g.*, points of a mug primarily concentrated on the body rather than the handle) can be transformed into a sparse feature modeling problem with uniform distributions of local point pairs.

We transform local point pairs into probability distributions via bins. For translation voting, A circle is divided into bins to determine the center. For orientation voting, bins are evenly distributed in a Fibonacci sphere to count the votes. The final prediction of the tracked pose will emerge with the largest vote count. For optimization, we place each part in a zero-center coordinate before voting. The universal voting strategy for each rigid part at $t$-th frame is conducted as follows:

★ Translation Voting. Denote the center by $o$ with the input point cloud. Therefore, the translation voting can be regarded as voting for the center of the part $\mathcal{P}_t^{(k)}$ in the camera space for the $t$-th frame point cloud $\mathcal{P}_t$.

For each point pair $p_1$ and $p_2$, we estimate the following two offsets:

$$\mu_t^{(k)} = (o - p_1) \cdot d \quad (12)$$

$$v_t^{(k)} = \|o - (p_1 + \mu_t^{(k)}d)\|_2 \quad (13)$$

$$where \quad d = \frac{p_2 - p_1}{\|p_2 - p_1\|_2} \quad (14)$$

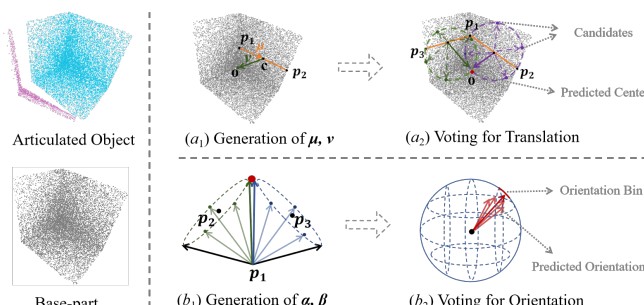

Articulated Object | ($a_1$) Generation of $\mu$, $v$ | ($a_2$) Voting for Translation

Base-part | ($b_1$) Generation of $\alpha$, $\beta$ | ($b_2$) Voting for Orientation

**Figure 4: Illustration for Translation voting scheme (Figure ($a_1$) and Figure ($a_2$)) and Orientation voting scheme (Figure ($b_1$) and Figure ($b_2$)). Please note that we conduct the voting scheme on each part but only one part is on demonstrating for simplicity's sake.**

As depicted in Figure 4 ($a_1$), once we get the key parameters $\mu_t^{(k)}, v_t^{(k)}$, we can determine the center with one degree-of-freedom ambiguity. To elaborate further, as depicted in Figure 4 ($a_2$) the object center will lie on a circle with its center $c$ and radius $\|o - c\|_2$ . During inference, divide the circle into bins to generate multiple votes. The bin with the most votes will be regarded as the final prediction.

★ `Orientation Voting.`
During the rotation process, the direction of upward and rightward exhibit invariance under SO(3), thus enabling voting for the prediction of these directions. To elaborate further, we vote for the upward and rightward direction in the camera space for all the parts of $\mathcal{P}_t$. The remaining direction is derived using the cross product. Mathematically, we define the upward orientation as $e_1$ and rightward orientation as $e_2$ (they are both unit vectors). Then we estimate the two angles using the point pair $p_1$ and $p_2$:

$$\alpha_t^{(k)} = e_1 \cdot \frac{p_2 - p_1}{\|p_2 - p_1\|_2} \quad (15)$$

$$\beta_t^{(k)} = e_2 \cdot \frac{p_2 - p_1}{\|p_2 - p_1\|_2} \quad (16)$$

Afterward, we conduct an elaborate partition with a Fibonacci sphere [14]. Bins can be evenly distributed on the sphere, which makes our orientation voting more accurate. As depicted in Figure 4 ($b_1$), once we get the key parameters $\alpha_t^{(k)}, \beta_t^{(k)}$, we can determine the orientation up to one degree-of-freedom ambiguity, which lies in a cone. During inference, we may generate multiple candidates with a constant degree interval around the sphere for each point tuple, demonstrated in Figure 4 ($b_2$). The final prediction will be the orientation with the most votes. Here, the orientation bin size is set to be $1.5°$.

★ `Scales.` Denoting the average bounding box scales as $\bar{\mathbf{s}}_t^{(k)} \in \mathbb{R}^3$ and the bounding box scale in a particular stream as $\mathbf{s}_t^{(k)} \in \mathbb{R}^3$, we predict the following statistic:

$$\gamma_t^{(k)} = \log(\mathbf{s}_t^{(k)}) - \log(\bar{\mathbf{s}}_t^{(k)}). \quad (17)$$

During inference, $\gamma_t^{(k)}$ is first averaged among sampled point pairs, and then the predicted scale $\hat{\mathbf{s}}_t^{(k)}$ can be retrieved using the function:

---

**Algorithm 1** Tracking algorithm with the universal voting scheme

1: **Input**: A live point cloud stream $\{\mathcal{P}_t\}_{t \geq 0}$, Per-part 9D pose $T_0^{(k)}$ at the first frame.
2: **Output**: Per-part 9D pose $T_t^{(k)}$ at all the $t > 0$ frames.
3: Initialize the reference state pool $B = \{\}$.
4: Add frame $t = 0$ into reference state pool $B$.
5: **for** $t > 0$ frames **do**
6:     **if** $t\%N == 0$ **then**
7:         Add $t$-th frame into reference state pool $B$.
8:     **end if**
9:     Obtain the nearest reference state $t'$ for $t$-th frame from reference state pool $B$.
10:     **for** $K$ rigid-parts **do**
11:         Pseudo-canonicalize the $k$-th part of cloud point $\mathcal{P}_t^{(k)}$ via the pose of reference state.
12:         Sample the point pairs and filter the noisy point pairs by part awareness.
13:         Predict the delta pose $\Delta T_t^{(k)}$ for the $k$-th part using the universal voting strategy.
14:         Compute the pose $T_t^{(k)}$ at current $t$-th frame.
15:     **end for**
16: **end for**

---

$$\hat{\mathbf{s}}_t^{(k)} = F(\gamma_t^{(k)}, \bar{\mathbf{s}}_t^{(k)}) = \exp(\gamma_t^{(k)}) \odot \bar{\mathbf{s}}_t^{(k)}, \quad (18)$$

where $\odot$ is the element-wise production.

**Loss Function.** During training, the $\mu_t^{(k)}, v_t^{(k)}, \alpha_t^{(k)}, \beta_t^{(k)}$ would be transformed into the corresponding probability distributions according to the bins used for voting. To measure the difference between the prediction and GT, we adopt the KL-divergence as the loss function, take $\mu_t^{(k)}$ for an example:

$$\mathcal{L}(\mu_t^{(k)}) = \sum_i \mu_{ti}^{(k)} \log\left(\frac{\mu_{ti}^{(k)}}{\hat{\mu}_{ti}^{(k)}}\right) \quad (19)$$

where $\hat{\mu}_{ti}^{(k)}$ is the prediction and $\mu_{ti}^{(k)}$ is GT($i$ refers to the $i$-th probability of the $i$-th bin in the distribution).

For the scale, we adopt the MSE loss as the objective function:

$$\mathcal{L}(\mathbf{s}_t^{(k)}) = \|\hat{\mathbf{s}}_t^{(k)} - \mathbf{s}_t^{(k)}\|_2 \quad (20)$$

The total loss for voting is the sum of all the loss functions above:

$$\mathcal{L}_{voting} = \sum_{k=1}^{K} \{\mathcal{L}(\mu_t^{(k)}) + \mathcal{L}(v_t^{(k)}) + \mathcal{L}(\alpha_t^{(k)}) + \mathcal{L}(\beta_t^{(k)}) + \mathcal{L}(\mathbf{s}_t^{(k)})\} \quad (21)$$

With all the voting methods above, we can easily obtain the $\Delta T_t^{(k)}$. Then the $t$-th frame pose can be obtained according to Equation 1. The overall articulation tracking procedure for the video is summarized in Algorithm 1.

**Table 1: Comparison with state-of-the-art on the synthetic dataset with the articulated objects from PartNet-Mobility.**

| Category | Method | Average Precision (%) ↑ | | | | | Inference Time (s)↓ |
| | | $3D_{25}$ | $3D_{50}$ | 5°
5 cm | 10°
5 cm | 15°
5 cm | |
|---|---|---|---|---|---|---|---|
| Laptop | A-NCSH [18] | 47.3, 35.3 | 29.7, 14.4 | 17.6, 8.3 | 33.2, 17.8 | 37.4, 24.2 | 1.67 |
| | OMAD [42] | 33.5, 27.3 | 18.9, 9.5 | 8.7, 5.3 | 23.8, 13.8 | 26.9, 15.8 | 0.34 |
| | Oracle ICP [46] | 37.6, 22.8 | 13.6, 10.0 | 7.9, 6.1 | 19.0, 12.6 | 24.8, 15.2 | 0.72 |
| | PA-Pose [26] | 59.8, 45.8 | 38.9, 23.6 | 27.6, 14.5 | 44.1, 30.2 | 49.3, 34.6 | 0.10 |
| | **VoCAPTER** (Ours) | **79.6, 58.3** | **58.3, 35.8** | **47.3, 24.6** | **63.2, 41.2** | **68.7, 46.3** | **0.07** |
| Eyeglasses | A-NCSH [18] | 41.8, 31.2, 30.8 | 29.8, 18.3, 17.3 | 14.8, 9.8, 9.3 | 23.9, 17.3, 16.5 | 32.9, 24.5, 23.6 | 2.59 |
| | OMAD [42] | 21.0, 19.8, 20.6 | 13.1, 11.2, 12.8 | 7.2, 5.5, 6.3 | 14.9, 12.6, 14.0 | 17.0, 14.1, 15.9 | 0.84 |
| | Oracle ICP [46] | 20.0, 17.8, 17.7 | 13.5, 10.3, 12.6 | 7.5, 5.8, 6.9 | 14.8, 11.7, 12.8 | 16.1, 12.8, 12.6 | 0.96 |
| | PA-Pose [26] | 51.3, 39.6, 38.3 | 41.3, 28.9, 27.8 | 32.4, 17.6, 16.9 | 42.3, 31.2, 30.8 | 45.8, 35.2, 34.2 | 0.14 |
| | **VoCAPTER** (Ours) | **65.3, 68.3, 65.8** | **52.1, 49.8, 50.3** | **41.1, 40.2, 39.6** | **56.8, 54.2, 54.1** | **61.3, 57.4, 57.8** | **0.09** |
| Dishwasher | A-NCSH [18] | 66.5, 47.5 | 52.2, 31.5 | 39.7, 22.3 | 56.1, 38.2 | 59.3, 42.1 | 1.70 |
| | OMAD [42] | 51.3, 35.6 | 36.2, 16.9 | 25.3, 8.5 | 40.4, 21.3 | 45.0, 23.8 | 0.36 |
| | Oracle ICP [46] | 47.6, 27.8 | 33.3, 16.8 | 20.8, 8.9 | 35.6, 14.3 | 40.6, 22.1 | 0.67 |
| | PA-Pose [26] | 84.1, 57.8 | 68.3, 42.1 | 55.3, 36.3 | 72.9, 48.3 | 76.3, 51.1 | 0.11 |
| | **VoCAPTER** (Ours) | **91.3, 76.8** | **74.3, 59.3** | **62.3, 47.3** | **78.6, 64.5** | **84.2, 68.0** | **0.06** |
| Scissors | A-NCSH [18] | 37.8, 36.3 | 24.3, 23.8 | 16.3, 14.5 | 23.7, 20.6 | 30.8, 29.2 | 0.05 |
| | OMAD [42] | 31.2, 33.5 | 18.3, 18.6 | 9.8, 10.1 | 17.3, 17.9 | 24.3, 25.6 | 0.29 |
| | Oracle ICP [46] | 27.8, 25.6 | 15.6, 13.8 | 7.8, 5.9 | 18.4, 15.7 | 21.3, 17.3 | 0.49 |
| | PA-Pose [26] | 43.6, 46.8 | 33.6, 31.3 | 24.6, 19.8 | 36.0, 33.9 | 39.1, 37.1 | 0.12 |
| | **VoCAPTER** (Ours) | **48.3, 50.1** | **38.2, 36.8** | **29.8, 28.3** | **40.3, 39.8** | **42.5, 42.2** | **0.08** |
| Drawer | A-NCSH [18] | 70.1, 66.3, 61.9, 65.8 | 54.3, 51.3, 49.6, 50.8 | 42.3, 40.1, 38.9, 41.6 | 51.7, 49.2, 46.3, 49.4 | 61.2, 57.4, 54.4, 57.1 | 3.64 |
| | OMAD [42] | 61.5, 58.3, 55.6, 58.6 | 47.9, 42.3, 43.2, 41.2 | 38.2, 31.3, 32.1, 32.2 | 49.8, 46.3, 45.3, 45.7 | 53.9, 51.0, 49.3, 51.0 | 0.62 |
| | Oracle ICP [46] | 55.9, 56.8, 52.3, 51.8 | 42.3, 39.2, 36.6, 40.2 | 29.8, 30.1, 25.8, 29.9 | 44.9, 43.2, 40.8, 44.8 | 49.2, 46.9, 44.8, 47.3 | 1.03 |
| | PA-Pose [26] | 90.3, 81.3, 78.6, 80.3 | 75.3, 68.6, 64.1, 66.0 | 62.3, 54.3, 52.1, 53.5 | 77.3, 70.8, 68.8, 70.3 | 83.1, 74.2, 70.8, 72.8 | 0.25 |
| | **VoCAPTER** (Ours) | **94.5, 87.5, 83.6, 84.9** | **79.8, 72.3, 66.5, 71.3** | **65.3, 61.3, 53.2, 59.8** | **83.6, 77.3, 71.4, 75.0** | **86.9, 80.3, 74.0, 77.2** | **0.12** |

## 5 EXPERIMENT

### 5.1 Experimental Setup

**Datasets.** To train our VoCAPTER, we generate the corresponding datasets for training and validation following [22]. Concretely, we first conduct experiments on a synthetic articulated object tracking dataset with the objects from PartNet-Mobility [41]. Next, we generate a semi-synthetic dataset for the articulated object tracking task from ReArt-48 repository [24], which generates more than 50K frames for each category. To further validate the generalization capacity of our VoCAPTER in real-world scenarios, we test the model trained by RobotArm. Please refer to the supplementary materials for more details about the datasets.

**Metrics.** We use the average precision to conduct the comparison between different methods, which is reported under both intersections over union (3D IoU) and pose error (translation and rotation error). 3D IoU is calculated between the predicted and GT bounding boxes with a threshold $\delta$. $3D_\delta$ is used to represent the Average Precision (%) of over $\delta$% IOU. For pose error, we report the Average Precision (%) under (5°, 5 cm), (10°, 5 cm), (15°, 5 cm) following [19, 48].

**Implementation Details.** During data pre-processing, input point clouds are downsampled to 2,048 points, and objects in RGB-D images are cropped and projected into the point cloud as network inputs. We set the initial learning rate to 0.001 and utilize cosine learning rate decay during training, with a total of 100 training epochs. All experiments were conducted on four NVIDIA GeForce RTX 4090 GPUs with 24GB of memory.

### 5.2 Comparison with the State-of-the-Art Methods

In this section, we conduct experiments on the synthetic dataset containing the articulated objects from PartNet-Mobility [41]. Quantitative experimental results of our VoCAPTER are reported in Table 1. Compared with single-frame pose estimation methods such as those proposed by A-NCSH [18] and OMAD [42], VoCAPTER demonstrates a significant improvement in per-part pose tracking performance, as evidenced by the $3D_{25}$ and $3D_{50}$ Average Precision metrics. For instance, considering the category *Scissors*, VoCAPTER achieves Average Precision scores of **65.3**%, **68.3**%, **65.8**%, surpassing the scores obtained by A-NCSH [18] (41.8%, 31.2%, 30.8%) and OMAD [42] (21.0%, 19.8%, 20.6%) in the $3D_{25}$ Average Precision metric. In terms of inference time, VoCAPTER achieves the fastest speed, averaging only **0.08** seconds per frame, surpassing both A-NCSH and OMAD. While our method exhibits only a slight improvement in pose tracking compared to PA-Pose, it demonstrates superior real-time performance. Therefore, we conclude that VoCAPTER effectively implements the voting strategy and fully leverages it in the pose tracking task. Figure 5 shows the qualitative results. It is evident from the results that VoCAPTER closely approximates the GT. This can be attributed to the effectiveness of the universal voting scheme in accurately modeling the geometry of rigid parts, thus enhancing registration accuracy.

### 5.3 Ablation Study

In this section, we conduct various ablation studies on our VoCAPTER. Only the results of base part from the category *Dishwasher* are reported, which are conducted in the PartNet-Mobility dataset.

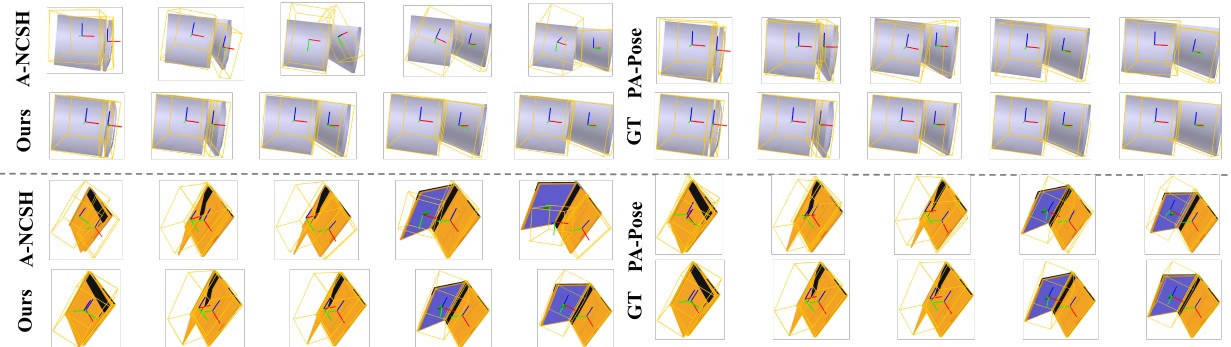

Figure 5: Qualitative results on PartNet-Mobility dataset. Two categories (*Dishwater* and *Laptop*) are shown.

**Table 2: Effects of point pair number and orientation bin size.**

| | Orientation Bin Size (°) | Average Precision (%) ↑ | | | | |
|---|---|---|---|---|---|---|
| | | $3D_{25}$ | $3D_{50}$ | 5° 5 cm | 10° 5 cm | 15° 5 cm |
| I | 1 | 90.9 | 74.1 | 61.3 | 78.6 | 83.9 |
| II (Ours) | 1.5 | 91.3 | 74.3 | 62.3 | 78.6 | 84.2 |
| III | 3 | 90.2 | 71.9 | 62.0 | 78.3 | 84.0 |
| | Awareness | Average Precision (%) ↑ | | | | |
| IV | - | 88.3 | 71.2 | 59.6 | 75.0 | 80.4 |
| V (Ours) | ✓ | 91.3 | 74.3 | 62.3 | 78.6 | 84.2 |

**Orientation bin size.** As mentioned in Section 4.4, bins are used for orientation voting and the bin size is set to be 1.5°. To verify the effects of orientation bin size, the ablation study results can be seen in Table 2 (I - III). From it, we infer that 1.5° is a better choice for orientation bin size, while larger or smaller bin sizes tend to introduce a big marginal error.

**Part Awareness.** With the aid of the part awareness mechanism, we filter the sampled point pairs for per-part pose tracking (*i.e.*, point pairs for per-part pose tracking should be from the target part meantime). Table 2 (IV - V) shows the ablation experiment results. It can be concluded that the proposed part awareness mechanism helps to achieve better performance. we conjecture this can be attributed to the elimination of noisy point pairs.

## 5.4 Generalization Capacity

**Experiments on Semi-Synthetic Scenarios.** We evaluate the effect of our VoCAPTER on the dataset ReArt-48 with semi-synthetic scenarios. Qualitative results are shown in Figure 6 (Top). The tracking results show that our method can perform well in the semi-synthetic scenarios.

**Experiments on Real-world Scenarios.** To investigate the tracking performance in real-world scenarios, we train and evaluate VoCAPTER on the 7-part RobotArm dataset [24]. Figure 6 (Down) shows the qualitative results. It is undeniable to suffer from the effect of the multi-depth structure of the robot arm instance.

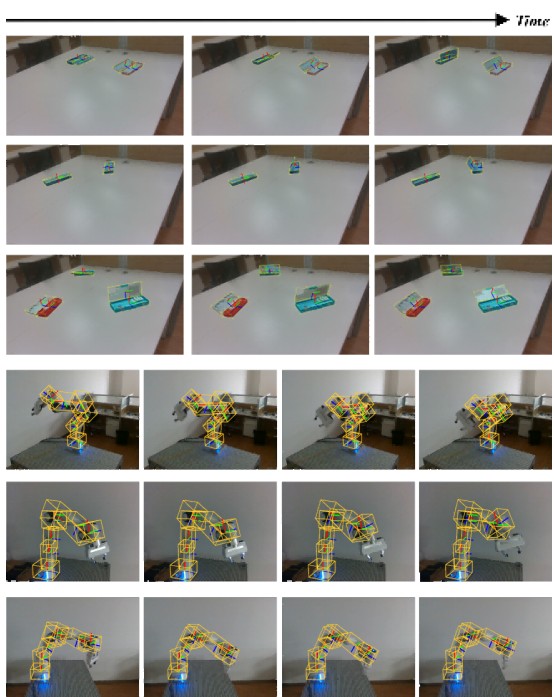

Figure 6: Demonstrations on semi-synthetic articulated objects (ReArt-48, Top). Qualitative results on real-world scenarios (RobotArm dataset, Down). Please zoom in for better visualization.

## 6 CONCLUSION

In this paper, we propose a novel framework, VoCAPTER, to conduct the category-level articulated object tracking task. Our method first leverages the inter-frame priors to conduct adjacent pose increment estimation task. Afterward, we perform the segmentation at per-part level and filter the noisy point pairs via proposed part awareness. Finally, we use the SE(3)-invariant parameters to vote for the pose at all t frames ($t > 0$). Empirical results demonstrate the superiority of our VoCAPTER compared to state-of-the-art methods not only on the synthetic dataset but also on real-world scenarios, which turns out to be a robust and real-time tracking framework.

## Acknowledgements

This work was supported by the National Natural Science Foundation of China under grant (No.32171888), the Dean's Fund of Hefei Institute of Physical Science, Chinese Academy of Sciences (YZJJ2022QN32), the Natural Science Foundation of Anhui Province (No.2208085MC57), and the National Key Research and Development Program of China (2019YFE0125700).

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
