# OpenReview forum: "VoCAPTER: Voting-based Pose Tracking for Category-level Articulated Object via Inter-frame Priors"
_acmmm.org/ACMMM/2024/Conference — MM2024 Poster_

### Official Review · Reviewer_pbmo · 2024-05-16

**Rating:** 5
**Confidence:** 3

**Summary:**

This research focuses on the problem of category-level joint object position tracking, especially for applications in multi-frame data streaming environments. The paper proposes a new pose tracking framework, VoCAPTER, by transforming the tracking task into an inter-frame pose increment estimation task and introducing SE(3)-invariant parameters for joint part pose voting. VoCAPTER utilizes a priori the motion relations between consecutive frames to update the position of the joint objects through a voting mechanism, providing a new approach to avoid direct pose regression. The experimental results demonstrate the effectiveness and generalization ability of the method in both synthetic and practical application scenarios. The proposal of VoCAPTER provides new perspectives and tools for the problem of joint object position tracking at the category level, and provides effective methodological support for research in this area.

**Strengths:**

1. Unlike traditional methods that track each frame independently, VoCAPTER utilizes a priori knowledge of continuous motion between frames. This approach not only improves the inference speed, but also improves the accuracy of the pose estimation by considering temporal coherence.
2. Using SE(3)-invariant parameters for component pose voting, VoCAPTER avoids the problems common to direct pose regression, which often struggles to deal with the nonlinearity and complexity of articulated object motion. This feature makes the model more robust to different orientations and motions.
3. the method introduces a probabilistic voting mechanism for pose estimation that effectively handles the uncertainty inherent in joint object tracking across sequential frames. This leads to better generalization and stability in pose prediction.
4. VoCAPTER not only performs well on synthetic datasets, but is also validated in real-world scenarios. This shows its practical applicability and robustness under diverse conditions.

**Limitations:**

1. The method's dependence on the inter-frame a priori and the voting mechanism may increase the computational overhead, which may affect its real-time applicability in resource-limited environments compared to simpler direct regression methods.
2. the performance of VoCAPTER relies heavily on the quality of the inter-frame motion information. In scenarios with drastic or highly unpredictable inter-frame variations, the accuracy of the system may be degraded.
3. while effective for the objects tested, it is unclear how the method can be extended to objects with more joints or more complex motion chains.
4. Although VoCAPTER shows good generalization ability in experiments, it is still a challenge to further verify its performance in more real-world scenarios .
5. When dealing with noisy or abnormal data, although some sensing mechanisms are introduced to improve the stability of prediction, how to ensure that it can still maintain high efficiency and accuracy in extreme situations is a problem that needs further research.

**Suitability:**

3

---

### Official Review · Reviewer_TEsa · 2024-05-23

**Rating:** 3
**Confidence:** 2

**Summary:**

This paper proposes VoCAPTER, a category-level articulated object tracking method based on inter-frame priors and pose voting strategy. VoCAPTER could conduct real-time tracking manner by utilizing inter-frame information and address ill-posed pose modeling problem by using a voting-based prediction. Experiments on the synthetic and real-word datasets demonstrate the effectiveness of the proposed method.

**Strengths:**

Better accuracy over baseline methods on synthetic datasets

**Limitations:**

1. Missing comparisons on the semi-synthetic and real-world datasets. Table 3 and Table 4 only contain experimental results of the proposed method, which lack comparison with other methods or any standards. Could the authors provide more comparisons or analyses on these datasets?
2. Some expression of the paper is not clear. Why voting-based prediction is better than regression-based prediction? In lines 165-166, I didn’t understand why the transformation from geometric features to probability features is “more friendly for learning”?
3. I am wondering why the authors didn’t compare the method with CAPTRA, which also could conduct 9DoF articulated object pose tracking?

Reference: Yijia Weng, He Wang, Qiang Zhou, Yuzhe Qin, Yueqi Duan, Qingnan Fan, Baoquan Chen, Hao Su, Leonidas J. Guibas, CAPTRA: CAtegory-level Pose Tracking for Rigid and Articulated Objects from Point Clouds, ICCV, 2021.

**Suitability:**

2

---

### Official Review · Reviewer_9rb5 · 2024-05-25

**Rating:** 3
**Confidence:** 1

**Summary:**

This paper addresses pose tracking for category-level articulated objects. It proposes a voting-based inter-frame incremental pose estimation method using decomposed features based on Category-based Point Pair Feature (CPPF).

**Strengths:**

- The method incorporates voting-based pose tracking, enabling efficient and accurate pose estimation.
- Comparative experiments on three datasets, PartNet-Mobility, ReArt-48, and RobotArm, show that the method outperforms the existing methods.

**Limitations:**

- The main concern is that the novelty of proposed method is unclear. The novelty of the canonicalization component seems limited, as it is a common practice to use the displacement from the previous frame. The part segmentation component uses PointNet++. A detailed explanation of the part awareness component needs to be included. The pose voting component is a minor tweak of existing methods Beyond PPF and CPFF.
- Figure 4 ($a_2$) and ($b_1$) are misleading. In Fig. 4($a_2$), points $p_1$ and $p_2$ are not necessarily on the circle. In Fig. 4 ($b_1$), points $p_2$ and $p_3$ should be at the centers of the circles, as in Figure 2 in the BeyondPPF paper.
- Minor comments
    - I wonder why 9D pose estimation is necessary for the pose voting step. Isn’t 6D enough if there is no part deformation? I wonder how the mask features $F_2$ and $F(p_1, p_2)$ are fused? In l. 552, how do we calculate the center $\mathbf{c}$ and radius $\lVert \mathbf{o} - \mathbf{c} \rVert_2$ from $\mu_t$, $\nu_t$?
    - In l. 869, the details of the RobotArm dataset are missing. Reference [22] seems to be incorrect.

**Suitability:**

2

---

### Meta-Review · Area_Chair_jrnU · 2024-06-26

**Recommendation:** Accept (Poster)
**Confidence:** 3

**Metareview:**

The submission presents a voting-based pose tracking method for category-level articulated objects, demonstrating its effectiveness through experiments on both synthetic and real-world datasets. The proposed method shows notable accuracy improvements and robust pose estimation by leveraging inter-frame motion information and SE(3)-invariant parameters.

However, some concerns regarding the novelty of certain components and the clarity of explanations were noted by reviewers. Specifically, the canonicalization and part segmentation methods appear to follow existing practices, and the comparative analysis on real-world datasets requires further enhancement. Despite these issues, the paper provides a solid foundation and offers a valuable contribution to the field.